# Transmission Line Object Detection Method Based on Contextual Information Enhancement and Joint Heterogeneous Representation

**DOI:** 10.3390/s22186855

**Published:** 2022-09-10

**Authors:** Lijuan Zhao, Chang’an Liu, Hongquan Qu

**Affiliations:** 1School of Control and Computer Engineering, North China Electric Power University, Beijing 102206, China; 2School of Information, North China University of Technology, Beijing 100144, China

**Keywords:** contextual information enhancement, joint heterogeneous representation, transmission line, object detection

## Abstract

Transmission line inspection plays an important role in maintaining power security. In the object detection of the transmission line, the large-scale gap of the fittings is still a main and negative factor in affecting the detection accuracy. In this study, an optimized method is proposed based on the contextual information enhancement (CIE) and joint heterogeneous representation (JHR). In the high-resolution feature extraction layer of the Swin transformer, the convolution is added in the part of the self-attention calculation, which can enhance the contextual information features and improve the feature extraction ability for small objects. Moreover, in the detection head, the joint heterogeneous representations of different detection methods are combined to enhance the features of classification and localization tasks, which can improve the detection accuracy of small objects. The experimental results show that this optimized method has a good detection performance on the small-sized and obscured objects in the transmission line. The total mAP (mean average precision) of the detected objects by this optimized method is increased by 5.8%, and in particular, the AP of the normal pin is increased by 18.6%. The improvement of the accuracy of the transmission line object detection method lays a foundation for further real-time inspection.

## 1. Introduction

The stable operation of the transmission line significantly affects the stability of the national economy and people’s livelihood [1]. In the transmission line, there are numerous components, such as insulators, vibration dampers, pins, etc., with a small size, complex shape and various installation positions, which cause great difficulty for the traditional manual inspection [2,3]. Therefore, the unmanned aerial vehicle (UAV) with a high detection speed, efficiency and security has been widely used in transmission line inspection [4].

In the field of computer vision (e.g., image classification [5,6], object detection [7] and semantic segmentation [8,9]), CNN (convolutional neural network) is widely used as a basic framework because of its local perceptibility and translation invariance. In the last decades, lots of classical deep learning models based on CNN have been proposed, such as AlexNet, VGG [10], GoogleNet [11], ResNet [12], DenseNet [13], MobileNet [14], ShffuleNet [15], EfficientNet [16], ResNeSt [17], etc. Based on CNN, numerous methods were proposed for transmission line object detection. Li et al. [18] achieved the detection of insulator defects by adding ResNet to the backbone network of SSD. Zhao et al. [19] detected insulators under a complex background by cutting object images and using Faster R-CNN. Bao et al. [20] used the improved YOLO algorithm to detect abnormal vibration dampers on the transmission line. Tang et al. [21] detected the grading ring by changing the size of the convolution kernel in Faster R-CNN. Yang et al. [22] could identify the shockproof hammer by multi-scale fusion and depth separable convolution. The mentioned methods are aimed at the large-scale objects in a uniform background in the aerial images. However, in the transmission line, as there are many types of components with large-scale differences, complex connections, and spatial location relationships, the proportion of small-scale objects is particularly large. Almost all small-sized objects have few characteristics and are always obscured by the other large-sized object in the visual field. For the detection of small-scale objects, Zhao et al. [23] proposed a visual shape clustering network to construct a bolt defect detection model. Jiao et al. [24] proposed a context information and multi-scale pyramid network to detect bolt defects in the transmission tower. In the previous detection methods, the features of the small object were not fully extracted, and the connection and spatial location relationships among each component were also wasted, as well as the opportunity of auxiliary detection.

CNN mainly focuses on local feature extraction, but its global feature extraction capability is insufficient. Recently, transformer has gained much interest because of its advantage of a self-attention mechanism in modeling the global relationship. The self-attention mechanism is to capture the correlation of feature vectors at different spatial locations within an image. In the early exploration of the self-attention mechanism, in order to accomplish the target task, non-local operation [25] obtained the center point features by fusing the neighboring point features. DETR [26] implemented transformer-based object detection with an end-to-end format for the first time. Dosovitskiyet et al. [27] proposed visual transformer (VIT) with good performance in image classification, which could be directly applied in image block sequences. In order to replace ResNet in downstream tasks, the pyramid vision transformer (PVT) [28] was obtained by stacking multiple independent transformer encoders and introducing the pyramid structure into the transformer. Although the pure transformer has an advantage in extracting global features, the extraction of the local features is ignored. Therefore, numerous studies have begun to focus on improving the ability of global feature extraction and strengthening the modeling of local information, such as transformer in transformer [29], Swin transformer [30], Regionvit [31], etc. Transformer in transformer [29] constructs a sequence of image blocks and a sequence of super-pixels by two transformer blocks, which could achieve the encoding of the internal structure information between pixels within a patch. In order to speed up the computation, Swin transformer [30] performed a self-attention mechanism by dividing windows; at the same time, the information interaction across could be realized by shifting windows. Regionvit [31] generated the region markers and the local markers in images with different patch sizes by using region-to-local attention, and the local markers received the global information by paying attention to the region markers. In sum, the self-attention in transformer focuses on the extraction of global features and ignores the contextual information interaction between two neighboring keys.

In this study, in order to improve the detection accuracy for small objects and multi-scale objects in the transmission line, an optimized model is proposed. The optimization includes two aspects: first, in the stage of feature extraction, the advantages of the Swin transformer in global feature extraction and the convolution in local feature extraction are combined. The weight coefficients of queries and keys with rich contextual information are obtained and then applied in the value with domain information. Second, in the stage of detection, the classification and localization features of the main representation are enhanced by the heterogeneous representation of the detected object, and the advantages of different representations are fully utilized. The detection accuracy of the optimized model is examined by an experimental dataset, and the results have shown that this model has good detection accuracy, especially for small-sized transmission line objects.

## 2. The Detected Object

At present, the data collection of the transmission line objects is mainly carried out by the UAV. According to the technical guide of the UAV inspection for the high-voltage transmission line, in the data collection, the minimum distance between the UAV and the transmission line is 10 m, while the maximum distance is determined by the performance of the UAV and the size of the detected objects. In this study, the scenes including the grading ring and the strain clamp are selected as the detected objects. As shown in Figure 1a, the grading ring scene contains the grading ring, stay wire double plate, adjusting plate and numerous pins with a small size. As shown in Figure 1b, the strain clamp scene contains the counterweight, stay wire double plate, strain clamp, grading ring and pins.

There are many factors that play a negative effect on the transmission line object detection. In particular, the visual field proportion of the detected object to the whole image is small, especially for small-sized objects, which are always seriously obscured, such as the pins and the grading ring in the red rectangle in Figure 2. The small-sized and obscured objects in the dataset increase the difficulty of detection.

The data in Figure 3a were obtained by a four-winged UAV with NIKON D90 and AF VR Zoom-Nikkor 80–400 mm f/4.5–5.6 D ED. Moreover, the amount and size of the various components in the transmission line are different, which causes the imbalanced category and the scale difference of the detected objects in the whole collected data. The statistics of the detected objects in the collected dataset are shown in Figure 3a. In the original dataset, the amount of the collected sample of the completed pin is the largest, followed by the counterweight, the adjusting plate, the stay wire double plate, the strain clamp, the grading ring, and the missing pin, which is the least. The size ratio of the detected object to the whole image is shown in Figure 3b. The size ratio of the grading ring is the largest, followed by the stay wire double plate, the strain clamp, the counterweight, the adjusting plate, and the pin, which is the smallest. The specific comparison is as follows: the size of the collected image is 4288 × 2848, the size range of the grading ring in the data is from 1170 to 2100, while the size of the pin ranges from 30 to 220. The size ratio of the pin is shown as a line, while the size ratio of the grading ring is shown as a block.

## 3. Method

In this study, we make full use of the object information in terms of both the implicit domain features and the explicit spatial location relationships in the feature extraction and the detection head, respectively. In the feature extraction, the contextual information feature is enhanced by fusing the convolution and the transformer. In the detection head, the classification and localization features are enhanced by weighting the object features of different representation types, and the heterogeneous features of different representations are fused to assist detection. The accuracy of object detection is improved from two aspects: the contextual information enhancement and the joint heterogeneous representations.

The framework of the object detection is shown in Figure 4. The Swin transformer enhanced by contextual information is used as the backbone of the network to extract features. In the process of feature fusion, FPN (feature pyramid network) [32] is used for feature fusion. In the detection head, on the basis of the original three branches of classification, regression and center-ness, a DVR (different visual representation) network is added to enhance the heterogeneous features (corner and center) of classification and localization features. The allocation of the positive and negative samples in the detection head is defined by combining the IOU of the predicted result and the anchor. In the network initialization stage, the IOU of the anchor is used a priori. In the training process, the IOU of the predicted result is added as a guide for the selection of the higher quality positive and negative samples so as to produce higher quality predictions.

### 3.1. Swin Transformer Architecture with Contextual Information Enhanced

In the backbone architecture, through the fusion of the self-attention key pairs for global information extraction and the convolution for local feature information extraction, the network possesses the representation ability of the global information and the local contextual information at the same time. The properties of the convolution and the self-attention are shown in Table 1.

Compared to the computation of CNN, the transformer is larger in computation. In this study, the input is a high-resolution image in order to reduce the computation as much as possible, and the self-attention computation of the non-overlapping windows in the Swin transformer is used as the backbone of the feature extraction. In the feature extraction, the role of the local receptive field in the shallow feature layer is much higher than that of the global receptive field, so the ability of the contextual information extraction is improved by the fusion convolution operation in the first stage. By adding the convolutional kernels into the similarity computation of key pairs and the self-attention output computation, it makes it such that the similarity computation of self-attention fuses the contextual information of neighboring ranges of key pairs, and the output global feature information contains the local receptive fields. Thus, the optimized model has a strong feature representation for small objects, and the network structure of the improved backbone architecture is shown in Figure 5.

The convolution operation is added to the self-attention calculation inside the MHSA of the local Swin transformer block in Figure 5. The convolution and the self-attention are generalized to a unified convolutional self-attention:(1)$$y_{i}=\underset{j\in L\left(i\right)}{\sum }\alpha_{i-j}\omega_{i-j}⊙x_{j}$$
(2)$$\alpha_{i-j}=\frac{e^{{\left(\omega_{q}x_{i}\right)}^{T}\omega_{k}x_{j}}}{\sum_{z\in L\left(i\right)}e^{{\left(\omega_{q}x_{i}\right)}^{T}\omega_{k}x_{z}}}$$
where *i, j*∈R are the indexes of the spatial location, *i* is the index of the kernel location, and *j*∈*L*(*i*) is the index of the local spatial neighborhood location of *i*. Taking a 3 × 3 convolution kernel as an example, *i* is index of the kernel location, *j* are the indexes included *i* and other eight indexes of the neighboring locations; *α_i_*_-*j*_∈(0,1) is the weight coefficient of each location in the summation; *ω_i_*_−*j*_ is the projection matrix of the spatial relationship of *i* and *j*; *x_j_* is the input feature vector; and *ω_q_* and *ω_k_* are the projection matrices of the query and the key, respectively.

Taking a 3 × 3 convolution kernel as an example, the unified convolutional self-attention model is shown in Figure 6; $\omega_{i-j}⊙x_{j}$ in Equation (1) is calculated by batch matrix multiplication (BMM). The generalized convolutional self-attention is expressed by a unified formula, and the calculation process of the model is adjusted by changing *α_i_*_−*j*_ and *ω_i_*_−*j*_ of the formula. In Figure 6a, the convolution kernel is applied in the input data by batch matrix multiplication, and the output, *y**_i_*, of a 3 × 3 matrix with the same value is obtained when the matrix *α_i_*_−*j*_ equals 1. In Figure 6b, through batch matrix multiplication, a dot product is obtained between the *ω_v_* and the input data, and the output of a 3 × 3 matrix is obtained by combining the weight coefficients of the matrix *α_i_*_−*j*_. The convolution calculation and the self-attention calculation have the advantages of the learnable filter and the dynamic kernel, respectively, which are unified by Equation (1). Therefore, the generalized convolutional self-attention incorporates the advantages of convolution and self-attention.

In the local Swin transformer block, the generalized convolutional self-attention is applied in the calculation process of the multi-head self-attention. The structure of the local enhanced convolutional self-attention is shown in Figure 7. The generalized convolutional self-attention is used to replace the self-attention. The weight coefficients of the query and the key with a 3 × 3 convolution are calculated firstly, and then, according to the weight coefficients, a weighted sum is obtained for the value that adds a 3 × 3 convolution, and finally, the feature fused with the contextual information is obtained.

### 3.2. Heterogeneous Representation Auxiliary Detection

In the field of object detection, some methods use the bounding box as the final output, which is the dominant representation, while the other methods use corner points and center points of the bounding box as the auxiliary representation. In this study, the bounding box, center point and corner point representation are summarized in terms of heterogeneous representations. The bounding box is easier to align with annotations, the center point representation can avoid many redundant boxes from a large number of anchors and is friendly to small object detection, while the corner point representation possesses a high localization accuracy. In order to combine the advantages of the mentioned heterogeneous representations into one framework, a DVR (different visual representation) network module is added to the detection head, as shown in the green box in Figure 4. The DVR module calculates the corner and center points through a small network, and then uses the object features represented by the center and corner as the auxiliary detection to enhance the classification and localization tasks of the original network. The calculation of this module is inspired by the idea of transformer, which uses the feature of heterogeneous representation as a key to assist in enhancing the main representation feature, so as to obtain the classification and localization feature assisted by heterogeneous representation. The feature enhancement is expressed as
(3)$$f_{i}^{’q}=f_{i}^{q}+\sum_{j}S\left(f_{i}^{q},f_{j}^{k},g_{i}^{q},g_{j}^{k}\right)\cdot T_{v}\left(f_{j}^{k}\right)$$
where, $f_{i}^{q},f_{j}^{’q},g_{i}^{q}$ are the input, output and geometric vectors of the query instance *i*, respectively; $f_{i}^{k}$ and $g_{i}^{k}$ are the input and geometric vectors of the key instance *j*, respectively; *T_v_*(·) is the value obtained by linear transformation; and *S*(·) is the similarity function calculation of *i* and *j*.
(4)$$S\left(f_{i}^{q},f_{j}^{k},g_{i}^{q},g_{j}^{k}\right)=softmax_{j}\left(S^{A}\left(f_{i}^{q},f_{j}^{k}\right)+S^{G}\left(g_{i}^{q},g_{j}^{k}\right)\right)$$
where$\;S^{A}\left(f_{i}^{q},f_{j}^{k}\right)$ is the similarity in appearance between *i* and *j*; and $S^{G}\left(g_{i}^{q},g_{j}^{k}\right)$ is the geometric term of the relative positions of *i* and *j*. The geometry vector of query is a 4D bounding box, and the geometry vector key is a 2D point (center point or corner point), which are obtained from the bounding box representation of the main network and the DVR network, respectively. When the dimension between the query geometry vector and the key geometry vector is different, convert the 4D bounding box into a 2D point, that is, the center point or corner point, and then calculate the similarity of the geometry vector between the key and the query.

## 4. Experiments

The original dataset is collected by UAV, and the useful data are selected manually. The dataset used in this study is constructed and expanded by way of rotating, changing the brightness and increasing the noise, where the preprocessing parameters are shown in Table 2. A total of 2950 images and 12,414 annotations of fittings and pins are obtained. There are seven types of objects: counterweight, grading ring, stay wire double plate, strain clamp, adjusting plate, normal pin and loss pin. The original image size is 4288 × 2848, and the sizes of the pin and the adjusting plate in the dataset range from 30 to 220 and from 50 to 400, respectively. The size ratio of the pin and the adjusting plate to the whole image is less than 0.12%, which is small in size, and can meet the definition of the relative size of small objects [33]. Therefore, in this study, the objects with the absolute size within 536 × 356 are referred to as small objects. The dataset is divided into the training set, validation set and test set, according to the amount ratio of 7:1:2. The number of classes in the three sets is not strictly at a 7:1:2 ratio. We counted the number of objects in these three sets, and the ratios of all objects are close to 7:1:2, which are listed in Table 3. The proposed model is trained and tested with a single NVIDIA GeForce RTX 2080Ti GPU, and the operating system is Ubuntu 20.04 with CUDA11.4 for training acceleration. Our experiment is implemented with PyTorch, and the batch size is set to 1 because of two aspects: first, the batch size is restricted by our hardware facilities, which only support a batch size of 1; second, the amount of the detected object in dataset is relatively small, and the convergence is good when the batch size is set as 1. The value of the epoch is proportional to the object diversity of the dataset. In this paper, the objects are less diverse, so the value of the epoch is relatively small. As shown in Figure 8, the value of the mAP of the experiment in this paper is almost a constant when the value of the epoch is larger than 39. Therefore, we set the value of the epoch as 48, which is enough to maintain the detection accuracy. We use stochastic gradient descent (SDG) for optimization. To avoid overfitting, we set the weight decay as le-4, and the learning rate as le-5.

### 4.1. Experiment Results

The original dataset of this paper contains high-resolution images. In order to minimize the pixel loss in the resized image and achieve good detection performance, a large-sized image is used in the experiment, and the input size of the original dataset is set to 2332 × 1400. In order to verify the detection accuracy of the optimized method as the input image size is reduced, the experimental image sizes are selected as 2332 × 1400, 2166 × 1300, 1999 × 1200, 1666 × 1000, and 1333 × 800. The corresponding learning times for these datasets are about 20 h, 18 h, 16 h, 13 h, and 10 h, respectively. The detection results are shown in Table 4; it can be found that when the image size increases from 1333 × 800 to 1666 × 1000, the total mAP is increased by 6.8%, and the AP of the normal pin is increased by 12.3%. When the image size increases from 1666 × 1000 to 1999 × 1200, the total mAP is increased by 7.8%, and the AP of the normal pin is increased by 14.3%. When the image size increases to 1999 × 1200, the increase in the mAP is small and is less than 1%. When the image size increases to 2332 × 1400 and 2166 × 1300, the total mAP is barely increased, and only the APs of the normal pin are increased by 2.6% and 1.2%, respectively. In general, with the increase in the input image size, the average detection accuracy of all objects is increased, especially for the small object of the normal pin. 

Figure 9 shows the visual results of the detected objects with different image sizes. Compared with Figure 9a–c, it can be found that with the increase in the input image size, the number of the detected normal pins on the parallel hanging plate (in the enlarged red circle) is increased, while the number of missed detections is decreased from 2 to 0. In Figure 9d,e, the obscured normal pin on the adjustment plate also can be detected. In Figure 9f, with the lowest image size, there is a false detection of the normal pin in the enlarged red circle. In Figure 9g, a missed detection of the normal pin located on the stay wire double plate exists. In Figure 9h,i, a normal pin at the same position is detected, and in Figure 9j, two adjacent normal pins at this position are accurately detected. Therefore, when the input image size is increased, the small and obscured objects can be successfully detected, and the detection performance is better. 

In sum, with the increase in the input image size, the detection accuracy is increased, but it is worth noting that the number of parameters (Para, measuring the spatial complexity) and the floating points of operations (FLOPs, measuring the computational complexity) are increased at the same time. Therefore, 1999 × 1200 is selected as the input image size in this study, which is based on the consideration of the calculation cost and the detection accuracy affected by the pixel loss.

The baseline method, dynamic ATSS, is compared with the optimized method based on the contextual information enhancement and joint heterogeneous representation (CIE-JHR) in this study. The comparison of the AP of single-type fittings are shown in Figure 10. Compared to the dynamic ATSS, the AP of the most fittings are improved by the CIE-JHR method, except for the adjusting plate object. The greatest increase in the AP is 18.6% for the normal pin, and the smallest increase in the AP is 1.5% for the loss pin. The AP of the counterweight, grading ring, stay wire double plate and strain clamp are increased by 2%, 2%, 3% and 4%, respectively. For the adjusting plate and loss pin, the AP is barely increased or is even decreased. The reasons are that the pin and the adjusting plate are small in size, and the samples of the loss pin and adjusting plate in the dataset are small. In addition, the loss pins are also easily confused with the normal pins, and the side view morphology of the adjustment plate is easily confused with the parallel hanging plate. In sum, these experimental results show that the optimized method can effectively improve the detection accuracy of almost all objects in the case of sufficient samples.

There are numerous methods for the detection of fittings and pins, such as SSD, Faster R-CNN, Retina Net, Cascade R-CNN, FCOS, and Dynamic ATSS. These methods with the same hyper-parameters are compared with the optimized method (CIE-JHR) in this study; AP50 (when IOU threshold is 0.5), Para and Time are used as the evaluation indicators. Para represents the number of model parameters in the training stage, and Time represents the detection time per image in the test stage. As shown in Table 5, the total mAP of the detected objects by CIE-JHR method is the largest, and the next are the Dynamic ATSS, FCOS, Cascade R-CNN, Retina Net, Faster R-CNN, and that by SSD is the smallest. Therefore, the CIE-JHR method possesses the best detection performance. For the small-sized objects, such as normal pin and loss pin, compared to the SSD method, the AP of the former object and the latter object detected by CIE-JHR method is increased significantly (62.8% and 41.7%, respectively). For the adjusting plate, the AP of CIE-JHR method is barely increased compared to the others methods, and is slightly decreased compared to the Dynamic ATSS method, the main reasons for which are explained in the preceding paragraph. 

The visual comparison of some typical object detection by the Dynamic ATSS method and CIE-JHR method are shown in Figure 11. Compared with Figure 11a,e, it can be found that the CIE-JHR method successfully detects the normal pin located on the adjustment plate, while the dynamic ATSS algorithm does not detect the normal pin. Comparison between Figure 11b,f show that the CIE-JHR method successfully detects the loss pin located on the strain clamp and excludes the mis-detected normal pin located on the front side of the stay wire double plate. Comparison between Figure 11c,j shows that the CIE-JHR method detects some small adjusting plates and counterweight, but there is a missed detection for the normal pin located on the stay wire double plate. Compared with Figure 11d,h, the CIE-JHR method detects the loss pin located on the strain clamp, while the Dynamic ATSS method fails to detect the loss pin. In summary, the CIE-JHR method has a good detection performance on small-sized objects, especially for the pin located on the adjusting plate, stay wire double plate and strain clamp. Therefore, the accuracy of object detection is successfully improved by adding the contextual information enhancement and the joint heterogeneous representation.

### 4.2. Ablation Analysis

In order to prove the effectiveness of the contextual information enhancement (CIE) of the feature extraction module and the joint heterogeneous representation (JHR) in the detection stage, ablation experiments are carried out based on the CIE-JHR method and the Dynamic ATSS method; the results are shown in Table 6. Compared to the pure Dynamic ATSS method, the AP, AP50 and AP75 of the Dynamic ATSS method with the CIE module are increased by 2.3%, 3.6% and 2.1%, respectively, which indicates that the feature extraction ability with CIE is significantly improved. The AP, AP50 and AP75 of the Dynamic ATSS method with the JHR module are increased by 1.8%, 3.5% and 2.7%, respectively, which indicates that the combination of the heterogeneous representations is conducive to the improvement of the detection accuracy. The AP, AP50 and AP75 of the CIE-JHR method are increased by 6.7%, 5.8% and 6.7%, respectively. These results show that the optimization of the contextual information enhancement in the feature extraction stage and the joint heterogeneous representation in the object detection stage can effectively improve the accuracy of the object detection.

## 5. Conclusions

In this study, a transmission line object detection method is proposed, which combines the contextual information and the joint heterogeneous representation. In this method, the Swin transformer is used as the backbone of the network structure, and the convolution is added in the self-attention calculation of the low-level structure, which can make full use of the contextual information. In the detection stage, the classification and regression tasks are carried out by combining heterogeneous representation features, which could improve the detection accuracy. The experimental results show that the total mAP of the detected objects by the optimized method is 67.1%, which is increased by 5.8% compared to the baseline algorithm, especially, the AP of the small-sized normal pins is 65.9%, which is increased by 18.6% compared to the baseline algorithm. It has a good detection effect for small objects obscured by other components and adjacent small objects with close distance. The optimized method can effectively improve the average accuracy of the transmission line object detection, which is beneficial to successfully detect the potential hazard of the small-sized and obscured objects in operation, and also can help to maintain the stable and safe operation of the transmission line. Moreover, the increase in the detection accuracy of the small-sized objects is also beneficial to the establishment of the efficient real-time detection of the transmission line object. In further study, the complex relationship between the transmission line hardware can be considered, and the artificial prior knowledge can be used to improve the object detection performance. 

## Figures and Tables

**Figure 1 sensors-22-06855-f001:**
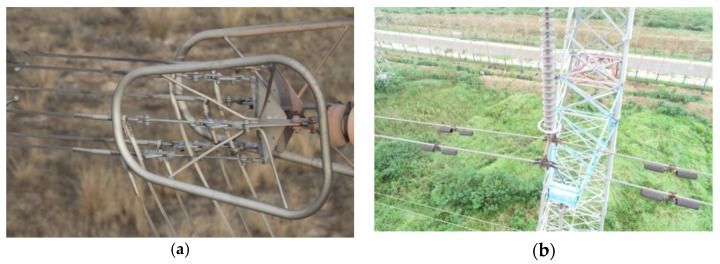
The detected objects. (**a**) The grading ring scene, (**b**) the strain clamp scene.

**Figure 2 sensors-22-06855-f002:**
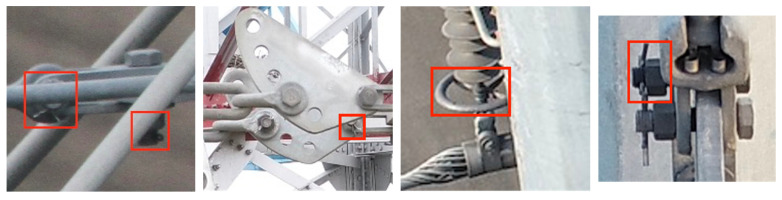
The small-sized and obscured fittings and connecting components.

**Figure 3 sensors-22-06855-f003:**
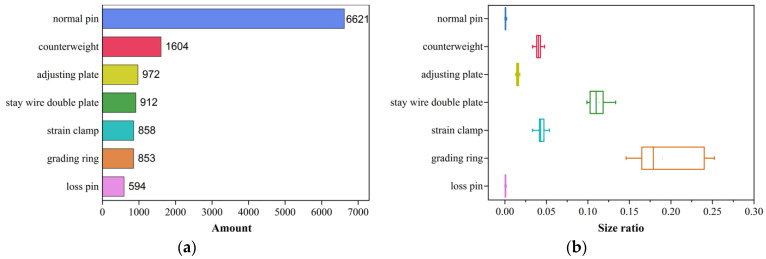
The statistics of the detected objects. (**a**) The amount of the detected objects, (**b**) the size ratio of the detected objects.

**Figure 4 sensors-22-06855-f004:**
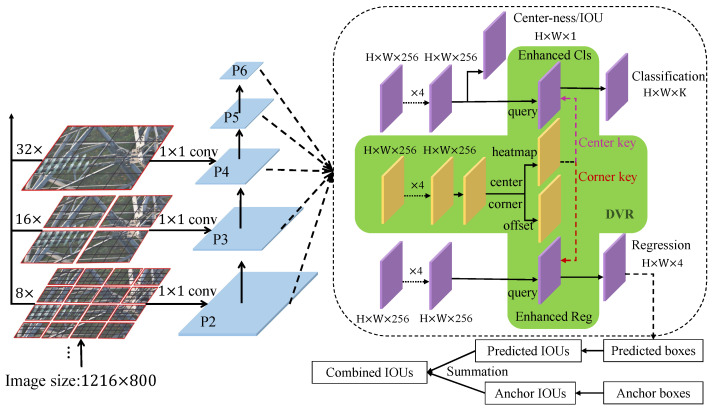
The framework of the heterogeneous representation auxiliary detection with contextual information. *H* and *W* are the height and width of the feature maps, respectively. *K* is the number of classes.

**Figure 5 sensors-22-06855-f005:**
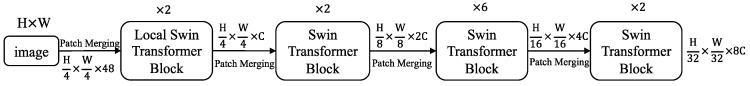
Swin transformer architecture with contextual information enhanced.

**Figure 6 sensors-22-06855-f006:**
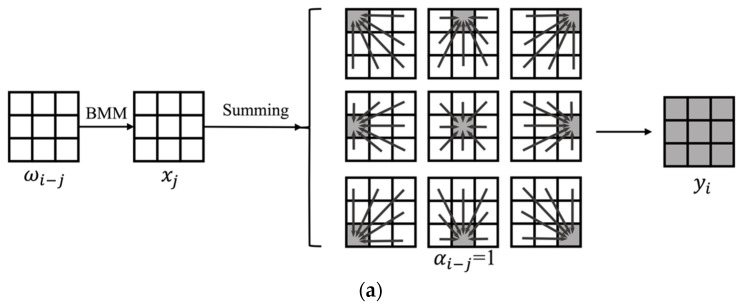
Illustration of the convolution and self-attention calculation. (**a**) The convolution calculation, (**b**) the self-attention calculation.

**Figure 7 sensors-22-06855-f007:**
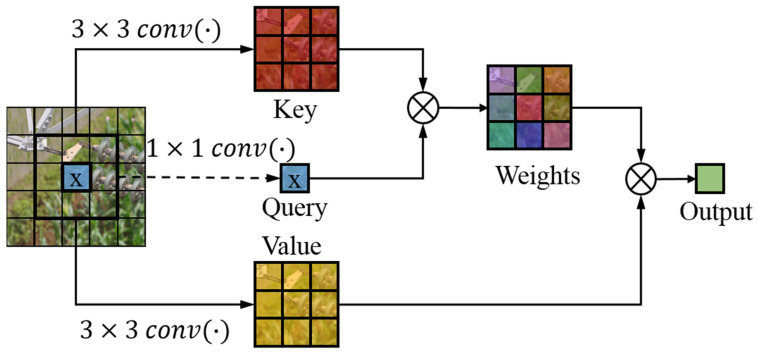
Contextual information enhanced convolutional self-attention module.

**Figure 8 sensors-22-06855-f008:**
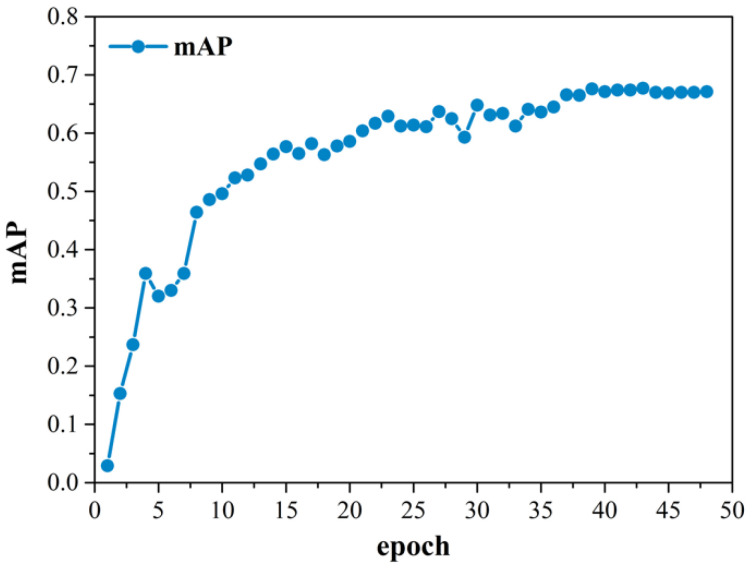
The mAP curve of the experiment in this paper when epoch is 48.

**Figure 9 sensors-22-06855-f009:**
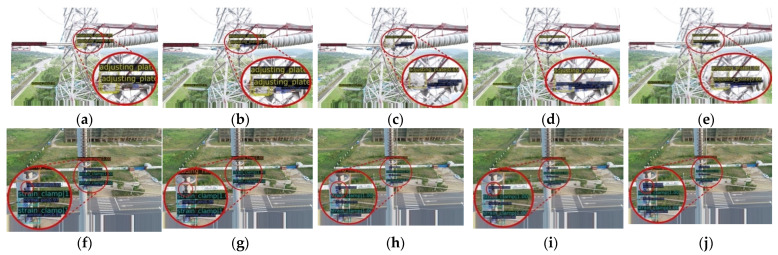
Visual results of the detected objects with different image sizes. (**a**) Size with 1333 × 800, (**b**) size with 1666 × 1000, (**c**) size with 1999 × 1200, (**d**) size with 2166 × 1300, (**e**) size with 2332 × 1400, (**f**) size with 1333 × 800, (**g**) size with 1666 × 1000, (**h**) size with 1999 × 1200, (**i**) size with 2166 × 1300, (**j**) size with 2332 × 1400.

**Figure 10 sensors-22-06855-f010:**
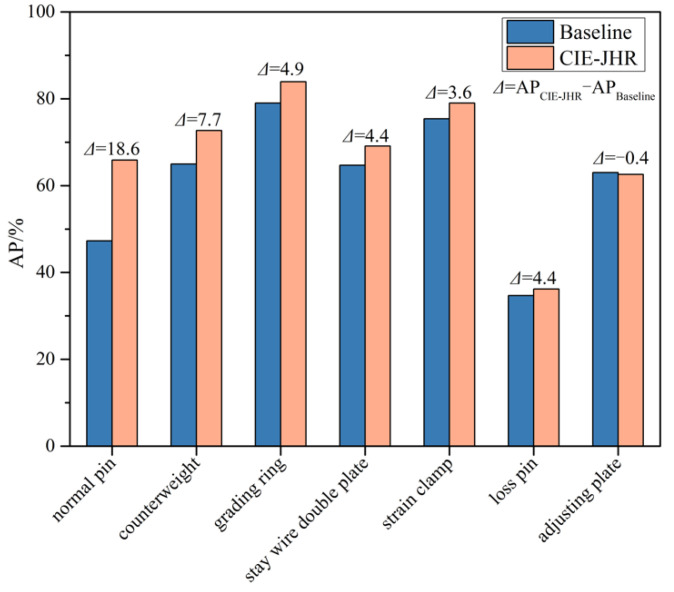
Comparison of the detected objects by CIE-JHR and dynamic ATSS methods.

**Figure 11 sensors-22-06855-f011:**
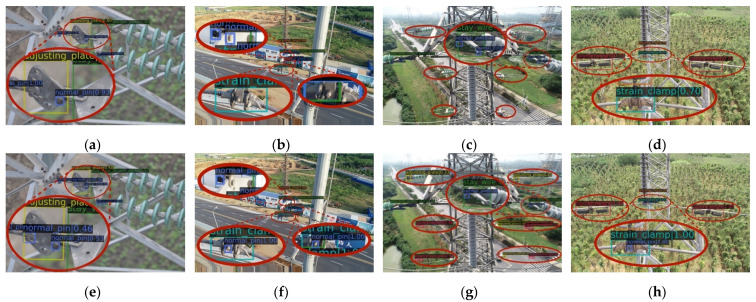
Visual results of the detected object by Dynamic ATSS and CIE-JHR method. (**a**) Insulator string and tower connection based on Dynamic ATSS algorithm, (**b**) strain clamp scene based on Dynamic ATSS algorithm, (**c**) multiple insulator string and tower connection based on Dynamic ATSS algorithm, (**d**) strain clamp and counterweight scene based on Dynamic ATSS algorithm, (**e**) insulator string and tower connection based on CIE-JHR method, (**f**) strain clamp scene based on CIE-JHR method, (**g**) multiple insulator string and tower connection based on CIE-JHR method, (**h**) strain clamp and counterweight scene based on CIE-JHR method.

**Table 1 sensors-22-06855-t001:** The properties of the convolution and the self-attention.

Property	Convolution	Self-Attention
Translation Equivariance	√	
Input-adaptive Weighting		√
Local Receptive Field	√	
Global Receptive Field		√

**Table 2 sensors-22-06855-t002:** The preprocessing parameters of the data augmentation.

Operation Name	Description	Range of Magnitudes
Rotate	Rotate the image magnitude degrees	[−30, 30]
Brightness	Adjust the brightness of the image. A magnitude = 0 gives a black image, whereas magnitude = 1 gives the original image.	[0.1, 1.9]
Noise	Increase Gaussian noise	Mean = 0, var = 0.001

**Table 3 sensors-22-06855-t003:** The class ratio of each object.

Object	Training Set	Validation Set	Test Set	Class Ratio
Normal pin	4654	675	1292	7.03:1.02:1.95
Counterweight	1139	153	312	7.1:0.96:1.94
Adjusting plate	672	99	201	6.91:1.02:2.07
Stay wire double plate	648	86	178	7.11:0.94:1.95
Strain clamp	594	79	185	6.92:0.92:2.16
Grading ring	605	82	166	7.09:0.96:1.95
Loss pin	431	53	110	7.26:0.89:1.85

**Table 4 sensors-22-06855-t004:** Comparison of the detected objects with different image sizes.

Input Image Size	Normal Pin	Counterweight	Grading Ring	Stay Wire Double Plate	Strain Clamp	Loss Pin	Adjusting Plate	mAP	Para/MB	FLOPs
1333 × 800	39.3	58.3	73.1	61.1	64.7	23.8	47.3	52.5	4126	185.03
1666 × 1000	51.6	65.1	77.2	65.2	72.3	29.6	54.1	59.3	5387	304.03
1999 × 1200	65.9	72.7	83.9	69.1	79	36.2	62.6	67.1	6644	425.24
2166 ×1300	68.5	72.8	84.3	69.2	79.3	36.5	62.7	67.6	7677	660.08
2332 × 1400	69.7	72.9	84.5	69.4	79.7	36.9	62.9	68	8534	782.54

**Table 5 sensors-22-06855-t005:** Comparison of the detection results.

Methods	Normal Pin	Counterweight	Grading Ring	Stay Wire Double Plate	Strain Clamp	Loss Pin	Adjusting Plate	mAP	Para /MB	Time /ms
SSD512 [34]	24.5	54.4	68.2	57.1	63.5	21.1	49.5	48.3	1754	87
Faster R-CNN [35]	41.5	60.1	73.2	58.2	69.3	25.9	56.2	54.9	2452	114
Retina Net [36]	43.5	61.4	76.2	61.1	71.5	31.6	60.5	58	2835	128
Cascade R-CNN [37]	45.7	64.8	77.1	62.7	72.6	31.5	61.4	59.4	5742	250
FCOS [38]	46.5	63.6	78	62.4	73.6	32.9	62.1	60	4372	107
Dynamic ATSS [39]	47.3	65	79	64.7	75.4	34.7	63	61.3	4642	120
CIE-JHR	65.9	72.7	83.9	69.1	79	36.2	62.6	67.1	6644	142

**Table 6 sensors-22-06855-t006:** The results of ablation experiments.

Method	Backbone	Head	AP	AP50	AP75
Baseline			35.6	61.3	37.4
	√		37.9	64.9	39.5
		√	37.4	64.8	40.1
CIE-JHR	√	√	42.3	67.1	44.1

## Data Availability

Not applicable.

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
