# Peer review of "Transmission Line Object Detection Method Based on Contextual Information Enhancement and Joint Heterogeneous Representation"

_sensors, 2022, doi:10.3390/s22186855_

Round 1

Reviewer 1 Report

Thanks to the author team for submitting the manuscript, the contributions made here are impressive. I have the will to accept it. However, here are some suggestions that may make the paper more complete, and I hope the authors can adopt it.

Comment 1: For your main work: CNN, Transformer and Object Detection. I would suggest adding some recent literature in the manuscript to flesh out your reference work. Some common examples include, preferably 2 or more:

1)      Commonality Autoencoder: Learning Common Features for Change Detection From Heterogeneous Images (10.1109/TNNLS.2021.3056238)

2)      Semi-Supervised Hyperspectral Image Classification via Spatial-Regulated Self-Training (10.3390/rs12010159)

Comment 2: In the abstract section, it is introduced that your proposed method is based on the Swin Transformer and optimized, and it is suggested that you can further clarify the CIE and HJR (give abbreviations), which can be more convincing.

Comment 3: The grammar and expression of some sentences can be improved, for example, "In the fields of image recognitions (e.g., object detection and semantic segmentation) (Line 35)", "self-attention (full text)", The first appearance of CIE-JHR should give the full name(Line 299)” etc. In addition, what is heterogeneous representations, (attention+ convolution) or (classification + regression)?

Comment 4: Some suggestions for improving the figures, I hope the author can adopt: 1: Where does the data in figure 3(a) come from, and what does the color lines in (b) mean?

2: Text note spacing in figure 4, "H × W".

3: The arrow at the end of figure 5 should point to some end process, not empty.

4: The ai-j in figure 7 seems redundant, and note the plural form of query-key-value.

Comment 5: Line 186, what does j in equation (2) mean, is i's neighbor point, and how to choose? Line 233, what is the geometry vector and how to get it?

Reviewer 2 Report

What is the definition of a small objects? In terms of image pixels, in terms of shooting angles, and so on. Because photos can be zoomed in to find small objects, but do they fit the definition of small objects in this article? That is, the scope of the research object in this paper needs to be particularly clear.

It is suggested that from the perspective of practical application, some suggestions need to be put forward to improve the effect for this paper.

Reviewer 3 Report

- The UAV was said to capture the figures from a safe distance. What are the minimum and maximum distances allowed between the UAV and the transmission line when collecting images? What exactly is meant by "safe distance"?

- The conclusion does not address the four difficulties mentioned in Section 2.

- It is difficult to understand the meaning of the following sentences: (i) "The amount of the various objects are complete pin > counterweight > adjusting plate > stay wire double plate > strain clamp > grading ring > missing pin." and (ii) "The scale of the various detected objects are grading ring > stay wire double plate > strain clamp > counterweight > adjusting plate > pin, the scale difference among these detected objects is large.", "As shown in Table 3, the total mAP of the detected objects is CIE-JHR method > Dynamic ATSS > FCOS > Cascade R-CNN > Retina Net > Faster R-CNN > SSD, CIE-JHR method possesses the best detection performance in general. ". What does it mean? 

- "FPN" was not defined anywhere. Kindly amend.

- In Figure 4. What is K ?

- Some of the figures includes mathematical notation, such as H and W. However, they are not defined anywhere prior to the figures. Kindly amend.

- What does BMM stand for?

- I would like to recommend the sub-caption for every figure to be written under the figure itself. For example, Figure 6(a) sub-caption (a) and (b) can be written under the figure itself.

- "The dataset used in this study is constructed and expanded by the way of rotating, changing the brightness and increasing the noise." Extensive information is required, such as the preprocessing parameters used..

- "The dataset is divided into the training set, validation set and test set according to the amount ratio of 7:1:2." Is the number of classes based on the same 7:1:2 ratio?

- "Our experiment is implemented with PyTorch, the batch size is set to 1, max training epoch is set to 48." Justification is required as to why those numbers are proposed?

Round 2

Reviewer 1 Report

The authors have revised the paper according to the comments, and this paper can be accepted in current form.

Author Response

We are very thanks to you for your final decision.

Reviewer 3 Report

The authors' efforts have been greatly appreciated. The following are additional feedback to further improve the manuscript:

1. Line 102-103, "the minimum distance between the UAV and the transmission line is larger than 10 meters". Minimum is a specific number. Do you 10 meters is the minimum distance between the UAV and the transmission line? "Minimum larger than 10 meters" is not the same as "Minimum 10 meters".

2. Line 380-382, "The experimental results show that the total mAP of the detected objects by the optimized method is increased by 5.8%, especially, the AP of the small-sized normal pins is increased significantly by 18.6%." Kindly report the actual mAP as well.

3. Authors response to Point 11 is not found in the latest manuscript. Kindly amend.
